# Protective Mechanism of *Polygonum perfoliatum* L. Extract on Chronic Alcoholic Liver Injury Based on UHPLC-QExactive Plus Mass Spectrometry Lipidomics and MALDI-TOF/TOF Mass Spectrometry Imaging

**DOI:** 10.3390/foods11111583

**Published:** 2022-05-28

**Authors:** Huaguo Chen, Lei Peng, Chao Zhao, Zongwei Cai, Xin Zhou

**Affiliations:** 1Key Laboratory for Information System of Mountainous Areas and Protection of Ecological Environment, Guizhou Normal University, Guiyang 550001, China; chenhuaguo@gznu.edu.cn (H.C.); penglei@gznu.edu.cn (L.P.); zhaochao@gznu.edu.cn (C.Z.); 2Guizhou Engineering Laboratory for Quality Control & Evaluation Technology of Medicine, Guizhou Normal University, 116 Baoshan North Rd., Guiyang 550001, China; 3State Key Laboratory of Environmental and Biological Analysis, Department of Chemistry, Hong Kong Baptist University, Hong Kong SAR 999077, China

**Keywords:** *Polygonum perfoliatum* L., chronic alcoholic liver injury, non-target lipidomics, mass spectrometry imaging

## Abstract

*Polygonum perfoliatum* L. has a long history of medicinal and edible applications. Studies have shown that it can significantly protect liver injury, but the mechanism is unclear. The purpose of this study was to explore the protective mechanism of *P**. perfoliatum* on chronic alcoholic liver injury from the perspective of lipid metabolism. After 8 weeks of alcohol exposure in male Wister mice, the levels of aspartate aminotransferase (AST), alanine aminotransferase (ALT) and alkaline phosphatase (ALP) in serum were significantly increased, and the activities of alcohol dehydrogenase (ADH) and acetaldehyde dehydrogenase (ALDH) in liver were significantly decreased. Meanwhile, pathological changes of liver tissue in mice were observed by histopathology. Then, Ultra-High Performance Liquid Chromatography (UHPLC) QExactive Plus Mass Spectrometer lipidomics and matrix-assisted laser desorption/ionization time-of-flight/time -of-flight (MALDI-TOF/TOF) mass spectrometry imaging methods were established to analyze lipid metabolism in mice. Ten different lipids were identified by statistical analysis, including Fatty Acyls, Glycerophospholipids, Prenol lipids and Sphingomyelins. After intervention with *P. perfoliatum* extracts at different doses (25 to 100 mg/kg), levels of AST, ALT, ALP in serum, and activities of ADH and ALDH in liver were significantly corrected. The hepatic cord structure was clear, and the liver cells were closely arranged without other obvious abnormalities. Non-target lipidomics analysis showed that *P. perfoliatum* extract could regulate the metabolic disorders of the 10 different lipids caused by continuous alcohol exposure. Pathway analysis suggested that the mechanism of *P. perfoliatum* extract on chronic alcoholic liver injury may be related to the regulation of linoleic acid and α-linolenic acid.

## 1. Introduction

Under the dual influence of fast-paced life and bad eating habits, the incidence of chronic liver disease represented by alcoholic liver disease (ALD) is increasing [1]. Data provided by the World Health Organization (WHO) shows that the prevalence of ALD in western countries is as high as 6% [2]. The global expenditure on ALD is increasing year by year, accounting for 5.1% of the total global disease burden [3]. Currently, the main treatment of ALD is a healthy lifestyle and alcohol withdrawal [4]. While the majority of patients are affected by social, environmental and personal influences [5], maintaining a healthy lifestyle to improve the effect of liver steatosis is not obvious, only 30% of patients can be cured [6]. Up to now, although most drugs for liver injury have prominent therapeutic effects, they have certain side effects and repeatability [7]. Therefore, it is the general trend to develop corresponding liver protection products [8], especially the products with natural products as raw materials [9].

*Polygonum perfoliatum* L. is a genus of *Polygonaceae*, which is distributed in China, South Korea, India and other Asian countries [10]. It grows in farmland, roadside and valley wetlands at an altitude of 80–2300 m [11]. *P. perfoliatum* can be processed into delicious dishes, high-quality livestock and poultry feed, and has high medicinal value [12]. Modern scientific research shows that it contains flavonoid and their glycosides [12], alkaloids, quinones [13], saponins, triterpenes, essential oil [10] and has the effects of anti-inflammatory [14], antitumor, antiviral [15] and antibacterial [10]. Meawhile, *P. perfoliatum* has a significant protective effect on liver injury [16], but the mechanism is still unclear, which seriously hinders its further development and utilization. Therefore, it is of great significance and value to carry out the research on the mechanism of liver protection for its development and utilization.

Lipidomics was first proposed in 2003 as one of the main branches of metabolomics [17]. It aims to study the lipid metabolism in body fluids, tissues and cells by various methods [18], explore the changes of lipid metabolism under different diseases or drug interference conditions, study the possible pathogenesis of diseases and the mechanism of drug action from the perspective of lipid metabolism network, and find key lipid biomarkers that can characterize diseases or drug intervention [19]. Recently, with the continuous development of efficient, high-throughput and high-sensitivity analysis methods [20], lipidomics technology has been widely used in the exploration of various liver diseases, especially in biomarker screening and pathogenesis research [21]. Thus, in the present study, in order to clarify the protective mechanism of *P. perfoliatum* on chronic ALD, lipidomics analysis was carried out and mass spectrometry imaging technology was used to verify the results.

## 2. Materials and Methods

### 2.1. Reagents and Materials

*P. perfoliatum* was taken from Guiyang, Guizhou Province, China and identified by Professor Sun Qingwen, Guizhou University of Traditional Chinese Medicine. Benzene datatum is purchased from Beijing Sun Pharmaceutical Co., Ltd. Bifendatatum were purchased from Beijing Taiyang Pharmaceutical Co. LTD. Alkaline phosphatase (ALP), alanine aminotransferase (ALT), aspartate aminotransferase (AST), acetaldehyde dehydrogenase (ALDH) and alcohol dehydrogenase (ADH) kits were acquired from Shanghai enzyme linked Biotechnology Co., Ltd. (Shanghai, China). High performance liquid chromatography grade methanol and acetonitrile were purchased from TEDIA company (Fairfield, CA, USA). Mass-grade formic acid was purchased from Roy Science Inc. (Charlotte, NC, USA). Ultrapure water was prepared using Millipore milliq purification system (Bedford, MA, USA), and all other chemicals were analytical.

### 2.2. Animal

Wister male mice were obtained from Changsha Tianqin Biotechnology co., Ltd. (Shanghai, China) (license No. scxk-2014-0011) which with a weight of 20 ± 2 g and were kept in animal room at temperatures of 20~25 °C and 60~70% of relative humidity. The animal experiments were conducted in accordance with the ‘Laboratory Animal Care and Use Guide (eighth edition)’ [22] and approved by the Animal Care and Use Committee of Guizhou Normal University. Project identification code: GZNU20210209, approval date: 18 February 2021.

### 2.3. Experimental Design

#### 2.3.1. The Herb Sample Preparation

Two hundred grams of *P. perfoliatum* was placed in a round-bottom flask and extracted with 500 mL distilled water for 3 h. The absorbent cotton was used to filter, the filter liquor was decompressed and concentrated, then freeze-dried, and the extract was obtained. Referring to the usage of human dosage of 10~15 g·d^−1^, as well as the guidance of pharmacology experimental methodology, a proper amount of the extract was precision weighing and using distilled water as solvent to prepare suspension with different concentrations.

#### 2.3.2. Control Drug Preparation

Bifendatatum tablets was converted to mouse dose at 0.5 g·kg^−1^ daily and was mixed with distilled water to form a suspension.

#### 2.3.3. Animal Grouping and Administration

After 7 days of basic feeding, 60 healthy male mice were randomly divided into 6 groups, namely blank control group (BCG), model control group (MCG), *P. perfoliatum* low dose group (PLG), *P. perfoliatum* medium dose group (PMG), *P. perfoliatum* high dose group (PHG) and Bifendatatum control group (PCG). Each group was given a fixed concentration of ethanol solution daily, except BCG, which was given an equal volume of normal saline (10 mL/kg body weight). The alcohol concentration increased by 5% from 20% (*v/v*) every three days to 40% final concentration, and remained 40% until 8 weeks [23,24]. Meanwhile, *P. perfoliatum* extracts and positive drug were used separately for intervention. The doses of PHG, PMG, PLG and PCG were 100, 50, 25, 50 mg/mL, respectively. Then, the mice were sacrificed at the end of the 8th week after intragastric administration. They were fasted for 12 h before execution and free to drink water. Blood was taken from the same eyeball vein of mice, placed at room temperature for 30 min, centrifuged at 1006.2× *g* for 15 min, carefully absorbed the upper light yellow serum and stored in separate containers. At the same time, liver tissues were collected and stored in −80 °C refrigerator.

### 2.4. Biochemical Index Determination

According to the manufacturer’ s instructions, the activities of AST, ALT and ALP in serum of mice were detected by corresponding detection kits. Ethanol dehydrogenase and aldehyde dehydrogenase kits were used to determine the activities of ADH and ALDH in liver homogenate.

### 2.5. Histopathological Analysis of Liver

After the mice were sacrificed, the tissues at the same position in the left lobe of the liver were immediately fixed with 4% neutral formaldehyde for 24 h. After ethanol gradient dehydration, paraffin embedding and HE staining, the prepared pathological sections were histologically characterized by light microscopy [25].

### 2.6. Lipidomics Analysis

#### 2.6.1. Extraction and Preparation of Lipid Samples

After the liver specimens stored at −80 °C recovered to 20 ± 2 °C, tissue samples (20 mg) taken from the same position in each liver were precisely weighed and placed into homogenizer tubes. Then, 200 μL of phosphate buffered saline (PBS) buffer was added, and the samples were homogenized. Next, 150 μL homogeneous mixture was transferred to a 1.5 mL tube, 1.2 mL pre-cooled methanol-methyl tert-butyl ether-water mixture (4:5:5, *v*/*v*) was added and vortexed for 30 s. The mixture was then placed on ice for 1 h and centrifuged at 16,000 rpm at 4 °C for 3 min [26].

Then, 100 μL of the lipid layer solution was transferred to a new tube and dried under nitrogen flow and stored in −20 °C refrigerator for use. Before injection into the LC-MS system, 1 mL of methanol-isopropyl alcohol (1:1, *v*/*v*) was added to enhance the resolution. After vortex mixing, the mixture was centrifuged at 16,000 rpm at 4 °C for 5 min, and 120 μL of supernatant was transferred to an injection bottle for analysis [27].

#### 2.6.2. Preparation of Quality Control (QC) Samples

A total of 10 μL of each redissolved sample was placed in a tube, fully vortexed and mixed evenly, and transferred to an injection bottle as a QC sample.

#### 2.6.3. Chromatographic Conditions

The samples were separated by Nexera LC-30 A UHPLC system. Mobile phase A was composed of acetonitrile and water with a ratio of 6:4, and contained 10 mM ammonium formate, while mobile phase B contained acetonitrile and isopropanol at a ratio of 1:9. The gradient elution procedure was: 0–2 min, mobile phase B maintained at 30%; 2–27 min, the mobile phase B changed from 30% to 100%; 27–35 min, the mobile phase B changed from 100% to 30%. Column temperature was 45 °C, flow rate was 300 μL/min, and injection volume was 3 μL. Throughout the analysis, the sample was stored in an automatic injector at 10 °C.

#### 2.6.4. MS Conditions

The samples were separated by UHPLC and analyzed using QExactive Plus Mass Spectrometer (Thermo Scientific^™^, Waldorf, Germany) electrospray ionization (ESI) in positive and negative ion modes. In positive ion mode, parameters of probe heater temperature, sheath gas flow rate, auxiliary gas flow rate, sweep gas flow rate, spray voltage, capillary temperature and S-Lens RF level were set at 300 °C, 45 arb, 15 arb, 1 arb, 3.0 kV, 350 °C and 50%, respectively. The scanning range of mass to charge ratio is 200 to 1800. In negative ion mode, related instrument parameters were set as 300 °C, 45 arb, 15 arb, 1 arb, 2.5 kV, 350 °C and 60%. The *m*/*z* scanning range for MS^1^ was 250 to 1800.

#### 2.6.5. Statistical Analysis

PeakView was employed to calibrate the MS peaks, subtract the background, normalize peak areas, and simplify the data. Markerview was used to generate a 3D matrix of sample names, mass charge ratios, retention times, and ion-pair signal strengths in positive and negative ion modes. After filtering noise, the matrix was imported into SIMCA-P 14.1 (The trial version, Umetrics, Umea, Sweden,) for standardization, and Hotelling’s T2 and DModX models were used to screen for abnormal samples. The principal component analysis (PCA) and orthogonal partial least squares discriminant analysis (OPLS-DA) functions of SIMCA-P 14.1 software (Trial version) were carried out to discriminate between the blank control and selected samples according to the operating procedures. The validity of the developed OPLS-DA model was evaluated based on R^2^Y (>0.5), Q^2^ (>0.5), response permutation testing (*n* = 200), and variable importance for the projection (VIP) values (>1.0). The lipid profiles having *p*-values < 0.05 in the statistical difference analysis and a VIP score ≥ 1 in the OPLS-DA were considered differential lipids. The lipid structure was identified by accurate mass number matching (error < 1 × 10^−5^) and secondary mass spectrometry analysis using databases including HMDB (http://www.hmdb.ca/, accessed on 20 November 2021), METLIN (http://metlin.scripps.edu/, accessed on 22 November 2021), and Lipid Maps (http://www.lipidmaps.org, accessed on 25 November 2021). In addition, the lipid identities were verified using the azo principle in positive and negative ion interaction mode [28].

### 2.7. MALDI-MSI Analysis

The mice livers were fixed on a cutting table with solid adhesive and sliced into 12-μm-thick samples using a CryoStar Nx70 cryostat (Thermo Fisher Scientific, Waldorf, Germany at −20 °C. The samples were then thawed and mounted on indium tin oxide (ITO)-coated glass slides. Before matrix spraying, the ITO slides were dried in a vacuum dryer for 20 min. Then, an appropriate amount of 2,5-dihydroxybenzoic acid (DHB) was dissolved in 70% methanol solution containing 0.1% trifluoroacetic acid (TFA) to yield a 20 mg/mL matrix solution. Following Wang’s [29] spraying protocol, an automatic substrate sprayer (ImagePrep, Bruker Daltonics, USA) was used to spray the DHB matrix onto the slices mounted on the ITO slides. The liver tissue sections sprayed with the matrix were analyzed using a Bruker Rapiflex MALDI-time-of-flight (TOF)/TOF mass spectrometer for MS-based imaging analysis. The instrument was equipped with a Smartbeam^™^ 3D laser at 355 nm with a sampling frequency of 5000 Hz and a laser energy setting of 65%. The MALDI MS imaging was performed in both positive and negative ion detection modes. The *m*/*z* range used for detection was 80–1000, and the spatial resolution for sample scanning was 100 μm. Data Analysis 4.0 was used for the preliminary treatment of the MALDI mass spectrometry data, and SCiLS Lab 2018b was used to construct the MS images.

## 3. Results

### 3.1. UHPLC-MS Analysis of the P. perfoliatum Extract

It is very important to know exactly what chemicals are in the extract for scientific understanding of its efficacy and mechanism. In the present study, an UHPLC-MS method was used to identify the main chemical components in the extraction of *P. perfoliatum*. The results are shown in Appendix A and Appendix A, 12 compounds were identified, including gallocatechin, catechinic acid, chlorogenic acid, epicatechin, sanitol-7-O-rhamnose-3-O-glucoside, rutin, kaempferol-3-O-neocyanin, isoquercitrin, kaempferol rutin, quercitrin, quercetin and kaempferol.

### 3.2. Activities Analysis of AST, ALT and ALP

The liver is a metabolic organ in the body and plays a role to oxidize, store glycogen and secrete proteins [30]. The liver also produces bile in the digestive system [31]. As an important part of human body, liver plays a vital role in human health. In this study, the main liver function indicators were investigated [32].

As shown in Table 1, compared with the blank group, the levels of AST, ALT and ALP in serum showed a significant increase trend, with statistical differences. On the one hand, this suggested that the modeling of this study was successful, and on the other hand, it also revealed that long-term exposure to alcohol did cause liver damage.

After continuous intervention of *P. perfoliatum* extract, abnormal AST, ALT and ALP levels were corrected. Compared with the model group, PHG, PMG and PLG had significant differences (*p* < 0.05), and showed a significant downward trend. There were no significant differences in PHG, PMG and PLG compared with the blank control group (*p* > 0.05). These results indicated that different doses of *P. perfoliatum* extract had a good repair effect on AST, ALT and ALP level disturbance caused by long-term exposure to alcohol.

### 3.3. Activities Analysis of ADH and ALDH

Liver is the main place of ethanol metabolism, ethanol oxidation system pathway, microsomal ethanol oxidizing system (MEOS) pathway and other major ethanol metabolic pathways can produce high toxic metabolites acetaldehyde [33]. After ethanol enters the liver, it is mainly metabolized to acetaldehyde by ADH and then further oxidized to acetic acid by ALDH [34]. When the blood ethanol concentration is too high, liver MEOS starts to decompose ethanol into acetaldehyde, which accelerates ethanol metabolism [35]. Clinical data showed that the activities of total ADH and ALDH in patients with alcoholic liver disease were lower than those in healthy subjects [36].

In this study, as shown in Table 2, the activities of ADH and ALDH in the liver of mice in the model group were significantly lower than those in the normal control group (*p* < 0.05), indicating that the activity of key enzymes of ethanol metabolism in liver of mice in model group may always be elevated under stress. Compared with the model group, the ALDH activity in the liver of mice in the high and medium dose groups of *P. perfoliatum* extract was significantly increased (*p* < 0.01), and the ALDH and ADH activities were almost reduced to the level of the normal control group. These results indicated that the intake of the extract of *P. perfoliatum* had a certain protective effect on the liver function of liver injury mice.

### 3.4. Effects on Liver Histopathology

Long-term excessive drinking can cause liver steatosis, fatty liver, and even evolved into alcoholic hepatitis and irreversible cirrhosis. As shown in Figure 1, under light microscope, the structure of hepatic cord in the blank group was clear, the liver cells were closely arranged, there was no lipid droplets in the cytoplasm, and no necrosis, inflammation and other pathological changes were observed. The model group had extensive swelling of hepatocytes, focal necrosis of hepatocytes, nuclear dissolution, enhanced eosinophilic cytoplasm, and more inflammatory cell infiltration, indicating that the mouse model of chronic alcoholic liver injury was successfully constructed.

Compared with the model group, the symptoms of liver cells in *P. perfoliatum* extract groups were improved, and the liver cord structure in the low dose group was clear, the cells were slightly swollen, and there was no obvious fatty degeneration in the tissue. The liver cells in the middle dose group were closely arranged, and a small amount of cells were slightly swollen, and there were no other obvious abnormalities in the tissue. The structure of hepatic cord in high dose group was clear, the liver cells arranged closely, and there were no other obvious abnormalities. The results of histopathological observation further showed that the extract of *P. perfoliatum* had a better effect on alcohol-induced liver cell injury in mice.

### 3.5. Analysis of Effects on Lipid Metabolism

#### 3.5.1. Quality Control of Non-Targeted Lipidomics Data

In this study, five QC samples were randomly inserted while 60 samples were detected to verify the reliability of the experimental method and the stability of the instrument. The unsupervised PCA analysis was performed on the data of the five QC samples after pretreatment and the results were shown in Figure 2.

The QC samples in positive and negative ion mode were all within 2std, indicating that the experimental method was reliable and the instrument was stable.

#### 3.5.2. Multivariate Statistical Analysis and Differential Lipid Identification

PCA is an unsupervised discriminant analysis, which can be used to judge grouping trends and outliers among groups. In this study, in order to observe whether there were significant differences in the overall situation of liver lipids in mice of different groups, the original data of each group were preprocessed, and the missing value and total peak area normalization were processed by MetaboAnalyst 5.0 online website. The processed data were analyzed by SIMCA 14.1 software for PCA multivariate statistical analysis.

As shown in Figure 3, the lipid data of liver tissue in the blank control group and the model control group were well separated, and the *P. perfoliatum* high, medium, low dose group and the positive drug group were near the blank group. It can be seen that there were significant differences in the differential lipids between the blank control group and the model control group, suggesting that long-term exposure to alcohol can cause lipid metabolism disorders in mice. The overall lipid level was close to the normal group after administration, indicating that *P. perfoliatum* has a good regulatory effect on lipid disorders caused by long-term alcohol exposure.

PCA is an unsupervised dimensionality reduction method that can effectively process high-dimensional data, but PCA is not sensitive to variables with small correlation, and PLS-DA can effectively solve this problem [37]. OPLS-DA combines orthogonal signals and PLS-DA to screen the difference variables, which is a correction of PLS-DA [38]. In the present study, OPLS-DA analysis was performed on the mass spectrometry data of blank and model group to further screen out the differential lipids. As shown in Figure 4A, in the negative ion mode, the fitting degree of the model R^2^X = 0.668 (cum) and the prediction ability of the model Q^2^ = 0.9 (cum) for the blank group and the model group, where R^2^ is greater than 0.5, the difference between R^2^X and Q^2^ is less than 4, and Q^2^ is close to 1, indicating that the model has good fitting and prediction ability in the negative ion mode. In positive ion mode (Figure 4B), R^2^X = 0.712 (cu), Q^2^ = 0.88 (cum), R^2^ > 0.5, the difference between R^2^X and Q^2^ is less than 3, and Q^2^ is close to 1, indicating that liver tissue also has good fitting and prediction ability in positive ion mode. In order to prevent the over-fitting of OPLS-DA model, 200 permutation tests were carried out for the establishment of OPLS-DA model, as shown in Figure 4C,D. The Y intercept of Q^2^ in positive and negative ion modes was less than 0, indicating that the OPLS-DA model did not over-fit, and the results were reliable.

Volcanic map combined Fold Change (FC) Analysis and Student’s *t*-test can intuitively show the aboriginality of lipid changes between the two groups [39]. With FC > 1.5 and Student’s *t*-test *p* value < 0.05 as the screening criteria, univariate analysis can intuitively show the aboriginality of lipid changes between the two samples, so as to help us screen potential marker lipids [40]. Figure 5 shows the volcano plot of the data in the blank group and the model group. It can be seen from the figure that there are obvious differences in lipids between the blank control group and the model control group, where the color blue indicates a down-regulation and the color red indicates an up-regulation, indicating that the lipid metabolites in the liver tissue of mice after long-term alcohol exposure have undergone great changes.

In order to determine the lipid abnormalities in mice with chronic alcoholic liver injury, variable importance for the projection (VIP) obtained by OPLS-DA model was greater than 1, Student’s *t*-test *p* value was less than 0.05, and FC > 2 or FC < 0.5 were used to screen differential metabolites. The lipids were identified by MS-MS spectra and database retrieval (https://www.lipidmaps.org, accessed on 21 February 2022), as shown in Table 3 and Appendix A, 10 different lipids were finally identified, including Fatty Acyls (FAs), Glycerophospholipids (PCs), Prenol lipids (PLs) and Sphingomyelins (SMs).

Cluster analysis was performed on the selected 10 differential lipid metabolites, as shown in Figure 6, red indicated the up regulation of lipid metabolites, and green indicated the downregulation of lipid metabolites.

It can be seen from the Figure 6 that the differential metabolites of model group and blank group were statistically different. The content of FAs, PCs and SMs in model group was lower than that in blank group, while the content of PLs was significantly higher.

#### 3.5.3. Regulatory Sffect of *P. perfoliatum*

In the present study, we found that the differential lipids in the liver of the blank group and the model group were significantly different, and further analyzed the lipids content under the intervention of the three doses of *P. perfoliatum* extract. The results showed that there were also significant differences in 10 different lipids between the intervention group and the model group.

As shown Figure 7 that all three doses of *P. perfoliatum* extract can correct lipid metabolism disorders in mice caused by chronic alcohol exposure, especially in the middle and high dose groups.

In order to verify the results of LC-MS lipidomics analysis, mass spectrometry imaging of differential lipids was systematically analyzed. As shown in Figure 8, highly abundant Stearoyl-L-Carnitine at *m*/*z* 424.3121, 6-Aminohexanoate at *m*/*z* 130.5262, Linoleic acid at *m*/*z* 279.4225, PC (16:0/18:1) at *m*/*z* 704.9583, LPE (20:3) at *m*/*z* 502.4112, SM (d18:1/18:0) at *m*/*z* 729.4961 and SM (d18:0/16:0) at *m*/*z* 704.3663 were located predominantly in peripheral region from the model group compared to the blank group. In contrast, down-regulated Bexarotene at *m*/*z* 347.7970, Ginkgolide B at *m*/*z* 425.8303 and Abietic acid at *m*/*z* 301.2924 were located in central regions. Meanwhile, the mass spectrometry images of the 10 different lipids in the livers of mice treated with 3 doses of *P. perfoliatum* extract were similar to those of the blank group, indicating that the disorder of lipid metabolism caused by chronic alcohol exposure has been corrected.

Furthermore, metabolic pathways of these 10 different lipids were also analyzed using the online software MetaboAnalyst 5.0 (https://www.metaboanalyst.ca/, accessed on 16 March 2022) in this study, and the results indicated that *P. perfoliatum* extract can treat chronic alcoholic liver injury by regulating the metabolism of linoleic acid and α-linolenic acid (Figure 9).

## 4. Discussion

In this study, we found the levels of AST, ALT and ALP in serum of mice with chronic alcoholic liver injury showed a significant increase trend, which revealed that liver function was impaired. Meanwhile, the activities of ADH and ALDH were significantly decreased. Histopathology analysis showed that damaged liver had extensive swelling of hepatocytes, focal necrosis of hepatocytes, nuclear dissolution, enhanced eosinophilic cytoplasm, and increased infiltration of inflammatory cells. Compared with long-term alcohol exposed mice, ALT, AST and ALP levels in *P. perfoliatum* extract group were significantly decreased, while ADH and ALDH activities were significantly increased. Histopathological examination showed that the liver cells in *P. perfoliatum* extract group had complete structure and clear morphology. The above results showed that the model was successfully established, and extract *P. perfoliatum* had a protective effect on mice with chronic alcoholic liver injury. Hepatic lipidomics studies showed that the intervention of *P. perfoliatum* extract had significant effects on linoleic acid and α-linolenic acid metabolism.

Linoleic acid (LNA) is a precursor of polyunsaturated fatty acids (PUFA) that can form longer N-6 fatty acids, often referred to as omega-6 fatty acids [41]. The distinguishing feature of omega-6 fatty acids is a carbon-carbon double bond on the sixth carbon of the methyl group [42]. Similarly, α-linolenic acid (ALA) is the precursor of n-3 fatty acid, known as-3 fatty acid. Its obvious characteristic is that there is a carbon-carbon double bond on the third carbon of methyl [43]. LNA and ALA are the basic dietary needs of all mammals, because they cannot be naturally synthesized in vivo, and they must undergo a series of similar transformations to form fatty acids [44].

After LNA enters the cell, it is catalyzed by acyl-coA 6-desaturase to gamma -linolenic acid (GLA). Then, through the extension of super-long-chain fatty acid protein 5, GLA was transformed into ditriglyceride (DGLA). Then, DGLA is transformed into arachidonic acid (AA) mediated by acyl coenzyme a (8-3)-desaturation enzyme.

Arachidonic acid is then converted to a series of transient metabolites known as icosadecanes, and finally to the final fatty acid form [45]. In the present study, linoleic acid levels in mice after chronic alcoholic liver injury were significantly increased, indicating that phospholipase hydrolysis of phospholipids releases hepatotoxic substances such as linoleic acid and secondary metabolites, which ultimately leads to liver cell death. After intervention with *P. perfoliatum*, the linoleic acid level in the liver of mice decreased significantly, suggesting that the protective mechanism may be related to the improvement of oxidative stress. Figure 6 showed that the α-linolenic acid content in model group was significantly higher than that in blank group, while the content was significantly decreased after *P. perfoliatum* intervention. Thus, in chronic alcoholic liver injury, phospholipase A (PLA) is activated, phospholipids are released by water to explain free linolenic acid, and *P. perfoliatum* plays a role in the treatment of acute liver injury by inhibiting PLA activity.

## 5. Conclusions

In this study, UHPLC-QExactive Plus mass spectrometry and MALDI-TOF/TOF mass spectrometry were used for non-targeted lipidomics analysis to study the protective mechanism of *P. perfoliatum* against chronic alcoholic liver injury. Multivariate statistical analysis showed that the establishment of alcoholic liver injury model was successful. Changes in lipid content suggested that chronic alcoholic liver injury could cause lipid metabolism disorder. After *P. perfoliatum* treatment, the lipids of 10 lipids were significantly restored to normal levels. Pathway analysis suggested that the protective effect of *P. perfoliatum* on liver might be related to the improvement of linoleic acid and α -linolenic acid metabolism.

## Figures and Tables

**Figure 1 foods-11-01583-f001:**
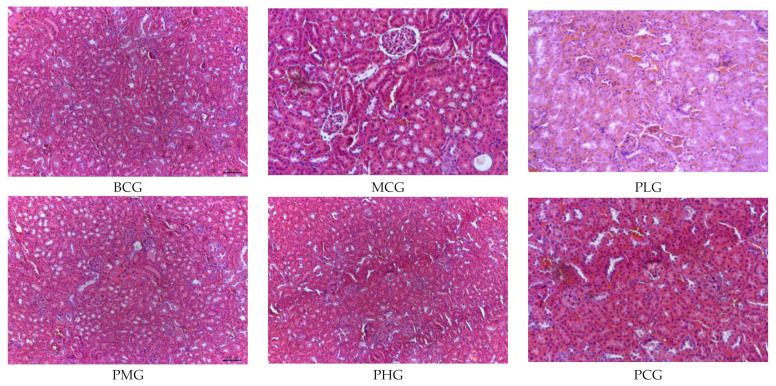
Effect of *P. perfoliatum* extract on liver HE staining induced by alcohol in mice (40×).

**Figure 2 foods-11-01583-f002:**
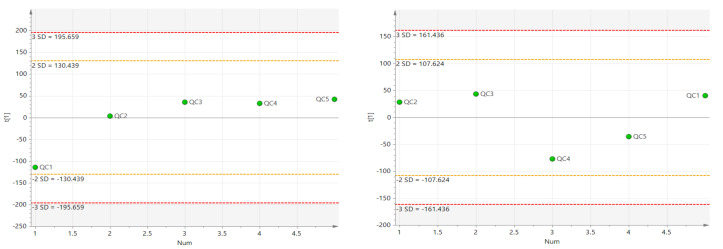
QC sample PCA score chart (Right: Negative ion mode; Left: Positive ion mode).

**Figure 3 foods-11-01583-f003:**
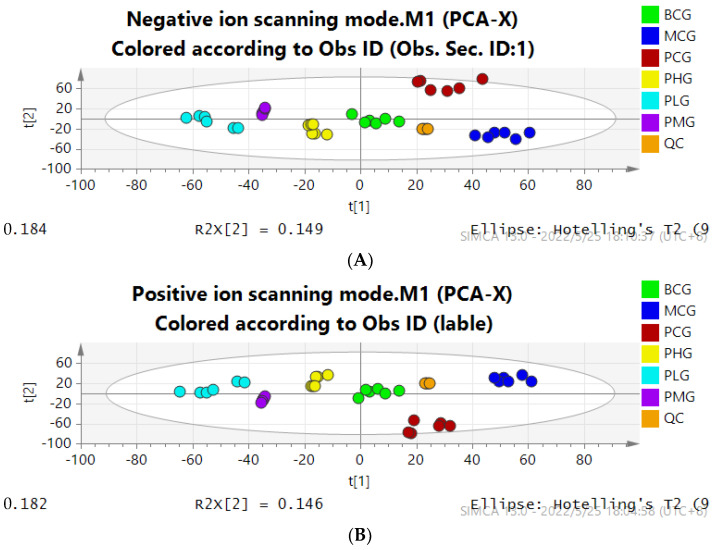
PCA score map of liver tissue in each group ((**A**): Negative ion mode; (**B**): Positive ion mode).

**Figure 4 foods-11-01583-f004:**
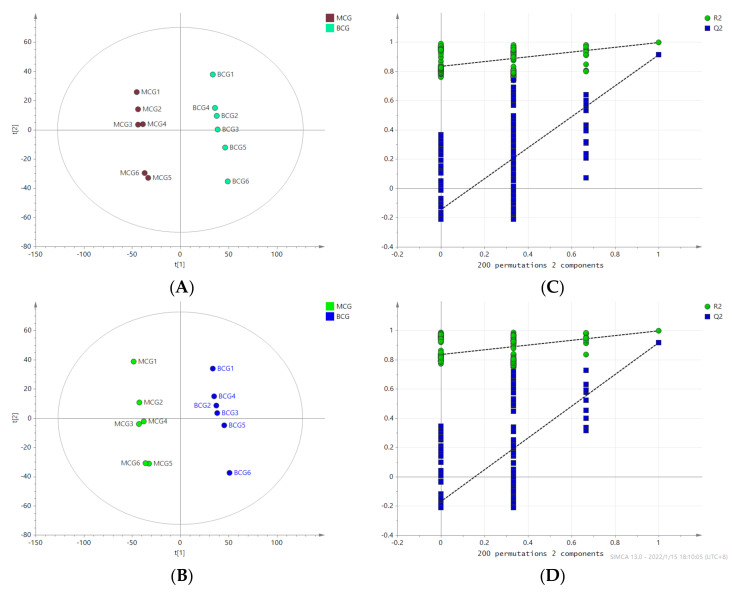
OPLS-DA score and OPLS-DA model response permutation verification map between blank control and model control group ((**A**,**C**): Negative ion mode, (**B**,**D**): Positive ion mode).

**Figure 5 foods-11-01583-f005:**
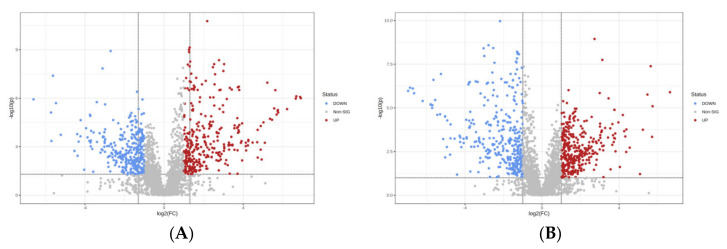
Volcano plot of OPLS-DA displacement test of blank control group vs. model control group ((**A**): Negative ion mode, (**B**): Positive ion mode).

**Figure 6 foods-11-01583-f006:**
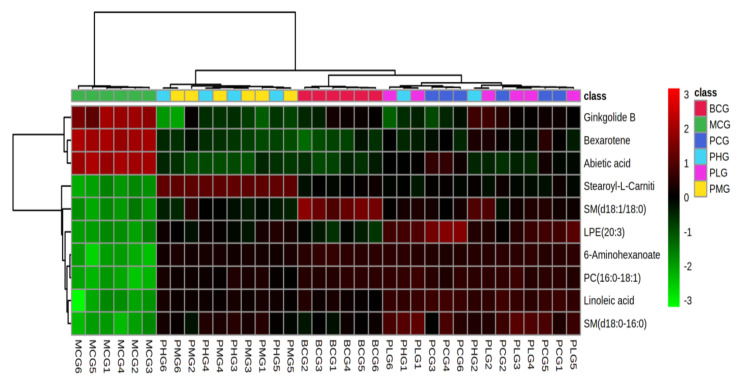
Heatmap visualization and hierarchical clustering analysis results of the differential lipids in each group.

**Figure 7 foods-11-01583-f007:**
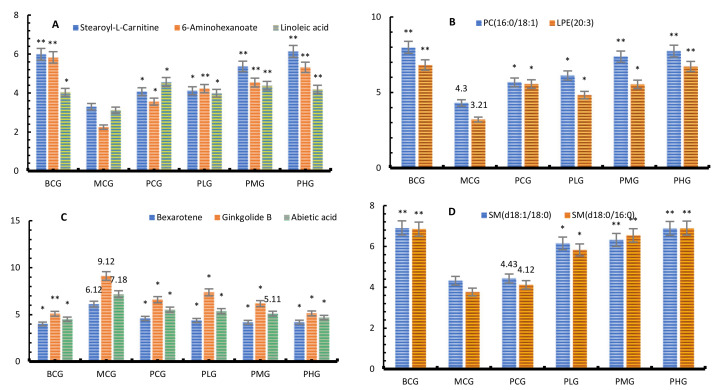
The relative content of 4 kinds of lipids ((**A**): FAs; (**B**): PCs; (**C**): PLs; (**D**): SMs). Note: compared with model group, * *p* < 0.05, ** *p* < 0.01.

**Figure 8 foods-11-01583-f008:**
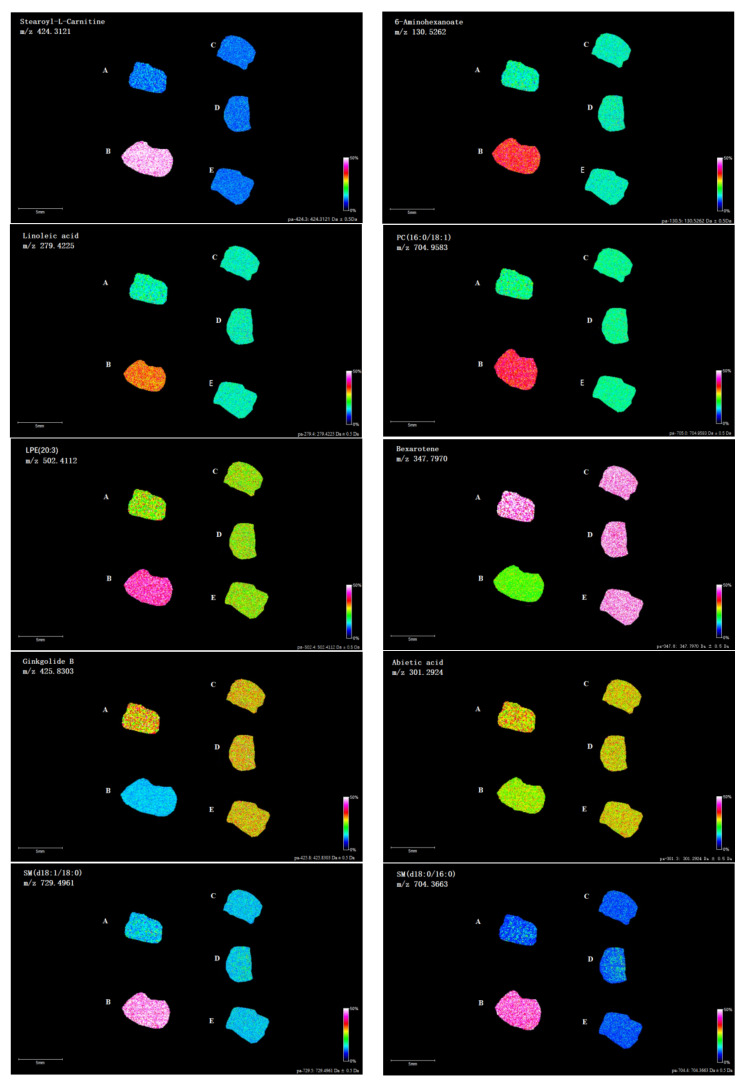
Mass spectrometry images of the 10 different lipids in the livers of mice (A–E were BCG, MCG, PLG, PMG and PHG groups, respectively).

**Figure 9 foods-11-01583-f009:**
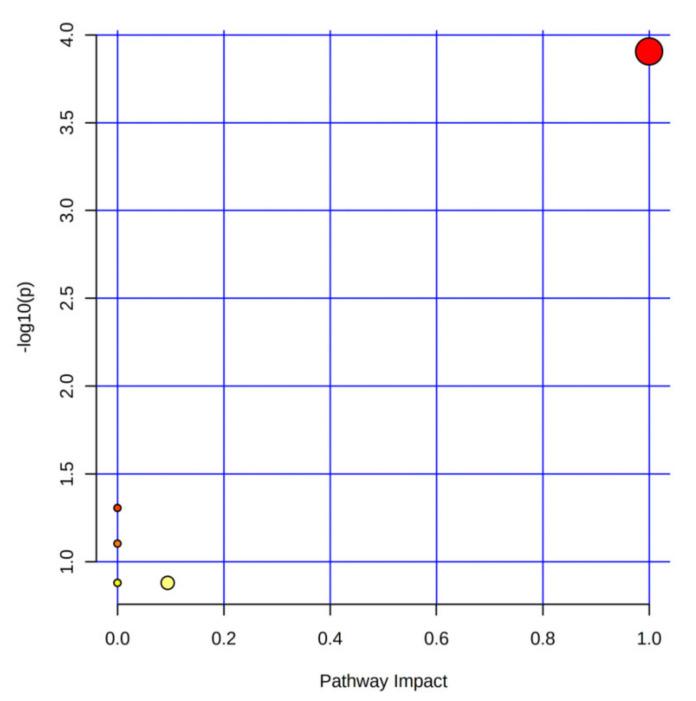
Pathway analysis of the differential lipids.

**Table 1 foods-11-01583-t001:** Liver function indicators of each group (n = 10, $\bar{X}$ ± s).

Group	ALT/µg·dL^−1^	AST/µg·L^−1^	ALP/ng·L^−1^
BCG	1.17 ± 0.11	1.06 ± 0.14	348.62 ± 13.61
MCG	1.89 ± 0.15 *	1.73 ± 0.12 *	457.11 ± 11.42 **
PLG	1.15 ± 0.12 ^#^	1.02 ± 0.16 ^#^	362.42 ± 21.23 ^##^
PMG	1.11 ± 0.16 ^#^	1.12 ± 0.13 ^#^	357.31 ± 24.41 ^##^
PHG	1.16 ± 0.18 ^#^	1.10 ± 0.08 ^#^	345.72 ± 18.89 ^##^
PCG	1.15 ± 0.10 ^#^	1.14 ± 0.14 ^#^	366.67 ± 19.61 ^##^

Note: compared with blank group, * *p* < 0.05, ** *p* < 0.01; compared with model group, ^#^
*p* < 0.05, ^##^
*p* < 0.01.

**Table 2 foods-11-01583-t002:** Effect of *P. perfoliatum* on key enzymes of ethanol metabolism induced by alcohol in mice (n = 10, $\bar{X}$ ± s).

Group	ADH Activity/(U•mg^−1^)	ALDH Activity/(U•mg^−1^)
BCG	1.23 ± 0.15	0.76 ± 0.11
MCG	0.81 ± 0.13 *	0.43 ± 0.07 *
PLG	1.15 ± 0.10 ^##^	0.72 ± 0.06 ^#^
PMG	1.51 ± 0.26 ^##^	1.01 ± 0.14 ^##^
PHG	1.66 ± 0.28 ^##^	1.20 ± 0.18 ^##^
PCG	1.44 ± 0.22 ^##^	1.13 ± 0.19 ^##^

Note: compared with blank group, * *p* < 0.05; compared with model group, ^#^
*p* < 0.05, ^##^
*p* < 0.01.

**Table 3 foods-11-01583-t003:** Detailed information of the different lipids between the blank group and model group.

NO.	Metabolite	Formula	Class	ESI Mode	Retention Time (min)	Measured *m*/*z*	MASS Accuracy (ppm)	Blank	Model
1	Stearoyl-L-Carnitine	C_25_H_49_NO_4_	FA	+	6.56	426.3421	−2.03	↑	↓
2	6-Aminohexanoate	C_6_H_12_NO_2_	FA	−	15.62	131.3162	−0.79	↑	↓
3	Linoleic acid	C_18_H_32_O_2_	FA	+	11.60	280.3325	−0.91	↑	↓
4	PC (16:0/18:1)	C_42_H_82_NO_8_P	PC	+	19.78	703.4683	1.49	↑	↓
5	LPE (20:3)	C_25_H_46_NO_7_P	PC	+	11.56	503.3612	−0.95	↑	↓
6	Bexarotene	C_24_H_28_O_2_	PL	+	23.45	348.5070	−0.71	↓	↑
7	Ginkgolide B	C_20_H_24_O_10_	PL	+	17.86	424.4803	1.35	↓	↑
8	Abietic acid	C_20_H_30_O_2_	PL	+	5.62	302.2624	−0.97	↓	↑
9	SM (d18:1/18:0)	C_41_H_83_N_2_O_6_P	SM	+	9.09	730.5561	−1.06	↑	↓
10	SM (d18:0/16:0)	C_39_H_81_N_2_O_6_P	SM	+	22.04	704.4763	−0.11	↑	↓

Note: ↑ indicates that the metabolite is up-regulated, ↓ indicates down-regulated.

## Data Availability

Data is contained within this article and Appendix A.

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
