# Peer review of "Protective Mechanism of Polygonum perfoliatum L. Extract on Chronic Alcoholic Liver Injury Based on UHPLC-QExactive Plus Mass Spectrometry Lipidomics and MALDI-TOF/TOF Mass Spectrometry Imaging"

_foods, 2022, doi:10.3390/foods11111583_

Round 1

Reviewer 1 Report

Comments:

Title

Polygonum perfoliatum author’s name could be included in the title. Please, use spectrometry instead of spectrometer.

Introduction

The authors could provide more details on phytochemical composition of the studied species.

Results and Discussion

The main disadvantage of the presented study is a lack of phytochemical investigation (concerning qualitative and quantitative analysis) of the studied Polygonum perfoliatum extract. The authors could applied UHPLC-MS for the analysis on the used extract.

The lipids identification should be more precisely described. Detailed comments on the MS/MS fragmentation of each identified compound should be presented. Is the authors have used comparison to authentic standards or?

Table 3. Numbers of C, H, and N should be in subscript.

Figure S1-S10. Please, use spectrum instead of spectrometry.

Rows 369-373. The text is in bold.

Author Response

Manuscript ID: foods-1718406

Title: Protective mechanism of Polygonum perfoliatum L. extract on chronic alcoholic liver injury based on UHPLC-QExactive Plus mass spectrometry lipidomics and MALDI-TOF/TOF mass spectrometry imaging

We are truly grateful for your critical comments and thoughtful suggestions. Based on these comments and suggestions, we have made careful modifications on the original manuscript and the revised version of the manuscript has been resubmitted to your journal. All changes made to the text are in red color. We hope the revised manuscript will meet the journal’s standard. Below you will find our point-by-point responses to the comments/questions, we sincerely hope to get your approval. If you have any comments/questions, please contact us, we’ll make serious changes.

Responses to reviewer 1:

1. Title:Polygonum perfoliatum author’s name could be included in the title. Please, use spectrometry instead of spectrometer.

RE: We corrected this error in the revised manuscript.

2. Introduction: The authors could provide more details on phytochemical composition of the studied species.

RE: In the revised manuscript, we added phytochemical composition information of Polygonum perfoliatum L..

3. Results and Discussion

The main disadvantage of the presented study is a lack of phytochemical investigation (concerning qualitative and quantitative analysis) of the studied Polygonum perfoliatum extract. The authors could applied UHPLC-MS for the analysis on the used extract.

RE: We do agree with your opinions and suggestions. In the revised manuscript, we added UHPLC-MS analysis of the extract in the section of “Results”.

4. The lipids identification should be more precisely described. Detailed comments on the MS/MS fragmentation of each identified compound should be presented. Is the authors have used comparison to authentic standards or?

RE: We do agree with your opinions and suggestions. In Table S2, we provided the MS/MS fragmentation of each identified compound. As you know, lipid reference substances are difficult to obtain, many or none, so this study did not adopt a standard comparison. In order to verify the accuracy of differential lipid screening, we used mass spectrometry imaging.

5. Table 3. Numbers of C, H, and N should be in subscript.

RE: In the revised manuscript, we corrected this error.

6. Figure S1-S10. Please, use spectrum instead of spectrometry.

RE: In the revised manuscript, we done it.

7. Rows 369-373. The text is in bold.

RE: In the revised manuscript, we corrected this error.

Reviewer 2 Report

The authors of the manuscript entitled "Protective mechanism of Polygonum perfoliatum extract on 2 chronic alcoholic liver injury based on UHPLC-QExactive Plus 3 mass spectrometer lipidomics and MALDI-TOF/TOF mass 4 spectrometry imaging". This original paper is interesting and could fall within the aim of the journal, but minor revisions are needed.

However some inaccuracies should be rectified, as indicated bellow and the revewed manuscript:

There are some inaccuracies in different part of the manuscripts, see bellow

- The scientific name of the plant should be written as Polygonum perfoliatum L.

- The full name for each abbreviation should be written correctly. For example Alcohol Liver    

     Disease “ALDA” at least one time in the text and the author should applicate this for all  

       breviations.

- Which animal the author used for his experiment, the rat or the mice????,

     that’s the question

_ It missed some references in section

  • (2.5 histopathogical analysis of liver)
  • (2.6.1 Extraction and preparation of lipid samples)

In section “3.3 Effect on liver histopathology”, the authors have to indicate in  the photo the effects of the extract of the plant on the liver, it is not obvious.

Author Response

Manuscript ID: foods-1718406

Title: Protective mechanism of Polygonum perfoliatum L. extract on chronic alcoholic liver injury based on UHPLC-QExactive Plus mass spectrometry lipidomics and MALDI-TOF/TOF mass spectrometry imaging

We are truly grateful for your critical comments and thoughtful suggestions. Based on these comments and suggestions, we have made careful modifications on the original manuscript and the revised version of the manuscript has been resubmitted to your journal. All changes made to the text are in red color. We hope the revised manuscript will meet the journal’s standard. Below you will find our point-by-point responses to the comments/questions, we sincerely hope to get your approval. If you have any comments/questions, please contact us, we’ll make serious changes.

Responses to reviewer 2:

1. The authors of the manuscript entitled "Protective mechanism of Polygonumperfoliatum extract on chronic alcoholic liver injury based on UHPLC-QExactive Plus  mass spectrometer lipidomics and MALDI-TOF/TOF mass spectrometry imaging". This original paper is interesting and could fall within the aim of the journal, but minor revisions are needed.

RE: We really appreciate your positive comments on the manuscript, as well as your questions and suggestions for revision. In the revised manuscript, we have made changes one by one, and hope that the revised manuscript can gain your approval.

2. However some inaccuracies should be rectified, as indicated bellow and the revewed manuscript:

There are some inaccuracies in different part of the manuscripts, see bellow

RE: We are very sorry for the inaccurate expression in the manuscript. In the modification of the manuscript, we have tried our best to correct it, and we hope to get your approval.

3. The scientific name of the plant should be written asPolygonum perfoliatum 

RE: We corrected this error in the revised manuscript.

4. The full name for each abbreviation should be written correctly.For example Alcohol Liver Disease “ALDA” at least one time in the text and the author should applicate this for all breviations.

RE: We corrected this error in the revised manuscript. In addition, the abbreviation and its full name are given for the first time in the manuscript.

5. Which animal the author used for his experiment, the rat or the mice????,that’s the question.

RE: Wister male mice were used for this experiment. In the revised manuscript, section “2.2. Animal ” provided the information. Meanwhile, We are very sorry for the clerical error in "2.7", “rat” should be “mice”.

6. How you can justify the use of these doses of acohol and thus the duration to induce chronic alcohol liver diseases?

RE: In this study, the model of chronic alcoholic liver injury and alcohol dose in mice were established after slight modification on the basis of references. In the revised manuscript, we added relevant references.

  • Bertola, A.; Mathews, S.; Ki, S. H.; Hua, W.; Gao, B. Mouse model of chronic and binge ethanol feeding (the NIAAA model). Nature Protocols2013, 8, 627–637. https://doi.org/ 1038/nprot.2013.032.
  • King, J. A.; Nephew, B. C.; Choudhury, A.; Poirier, G. L.; Lim, A.; Mandrekar, P., Chronic alcohol-induced liver injury correlates with memory deficits: Role for neuroinflammation. Alcohol 2020, 83, 75-81.https://doi.org/10.1016/j.alcohol. 07.005.

7. “BCG group was given intragastric administration of sterile distilled water with an intragastric dose of 10 mL/kg body weight. At the same time, 10 mL/kg body weight was given to mice by intragastric administration of the extracts and positive drugs every afternoon.”This sentence is very confusing,would you like to restructure it to make it clearer for the readers.

RE: We apologize for the lack of clarity. In the revised manuscript, we re-described it, hoping to gain your approval. In addition, we also carefully revised the description of other parts of the manuscript. The following is the revised description of this part.

“Each group was given a fixed concentration of ethanol solution daily, except BCG, which was given an equal volume of normal saline (10 mL/kg body weight). The alcohol concentration increased by 5% from 20% (V/V) every three days to 40% final concentration, and remained 40% until 8 weeks [24, 25]. Meawhile, P. perfoliatum extracts and positive drug were used separately for intervention. The doses of PHG, PMG, PLG and PCG were 100, 50, 25, 50 mg/ml, respectively. Then, the mice were sacrificed at the end of the 8th week after intragastric administration.”

8. The conditions of centrifiugation should be stated in g units instead of rpm units. Please modify it.

RE: In the revised manuscript, we revised this description.

9. It missed some references in section

(2.5 histopathogical analysis of liver)

(2.6.1 Extraction and preparation of lipid samples)

RE: In the revised manuscript, relevant references were added.

10. Please list the software you used to conduct these analyses, together with their licensing oinformation.

RE: In the revised manuscript, these informations were added.

11. In section “3.3 Effect on liver histopathology”, the authors have to indicate in the photo the effects of the extract of the plant on the liver, it is not obvious.

RE: In the revised manuscript, the relevant photos were corrected.

Round 2

Reviewer 1 Report

The authors have revised the manuscript according to the reviewer`s recommendations.